# SLTrain: a sparse plus low-rank approach for parameter and memory efficient pretraining

**Andi Han**[1]    **Jiaxiang Li**[2]    **Wei Huang**[1]    **Mingyi Hong**[2]    **Akiko Takeda**[1,3]
**Pratik Jawanpuria**[4]    **Bamdev Mishra**[4]
[1]RIKEN AIP  (andi.han@riken.jp, wei.huang.vr@riken.jp)
[2]University of Minnesota, Twin Cities  (li003755@umn.edu, mhong@umn.edu)
[3]University of Tokyo  (takeda@mist.i.u-tokyo.ac.jp)
[4]Microsoft, India  (pratik.jawanpuria@microsoft.com, bamdevm@microsoft.com)

## Abstract

Large language models (LLMs) have shown impressive capabilities across various tasks. However, training LLMs from scratch requires significant computational power and extensive memory capacity. Recent studies have explored low-rank structures on weights for efficient fine-tuning in terms of parameters and memory, either through low-rank adaptation or factorization. While effective for fine-tuning, low-rank structures are generally less suitable for pretraining because they restrict parameters to a low-dimensional subspace. In this work, we propose to parameterize the weights as a sum of low-rank and sparse matrices for pretraining, which we call SLTrain. The low-rank component is learned via matrix factorization, while for the sparse component, we employ a simple strategy of uniformly selecting the sparsity support at random and learning only the non-zero entries with the fixed support. While being simple, the random fixed-support sparse learning strategy significantly enhances pretraining when combined with low-rank learning. Our results show that SLTrain adds minimal extra parameters and memory costs compared to pretraining with low-rank parameterization, yet achieves substantially better performance, which is comparable to full-rank training. Remarkably, when combined with quantization and per-layer updates, SLTrain can reduce memory requirements by up to 73% when pretraining the LLaMA 7B model.

## 1  Introduction

Large foundation models have achieved tremendous success in various domains, including linguistics, computer vision and biology. In particular, large language models (LLMs), such as the GPT series [39, 5] and the LLaMA family [51, 52] have reshaped the perception of how machine understands human languages. The predominant success of these models is primarily due to the model size, usually scaling to hundreds of billions of parameters. The scaling laws seem to suggest the capacity of LLMs grows with the model size [25], but nonetheless requiring massive amount of resources for pre-training, storing, and fine-tuning. Particularly, memory requirement for training an LLM imposes a hard barrier for model deployment on commercial GPUs. For example, the LLaMA 7B model requires a minimum memory cost of approximately 42G under 16-bit floating point, including 14G of parameter state and 28G of optimizer state for momentum-based optimizers, like Adam [59, 28].

Building an LLM (from scratch) for downstream tasks typically involves two phases, i.e., pretraining and fine-tuning. The goal of pretraining is to capture general language patterns and semantics, enabling the model to acquire useful representations of words and sentences. Common pretraining objectives include masked language modeling [26], next token prediction [39, 40], etc.

38th Conference on Neural Information Processing Systems (NeurIPS 2024).

Fine-tuning then tailors the learned model representations from pretraining to downstream tasks, adjusting its weights to enhance performance on specific objectives. Pioneered by LoRA [21], recent works have popularized low-rank finetuning of a given pretrained model ($W_0$), where $W_0$ is generally full-rank (i.e., pretrained without any constraints). The premise is that LLMs usually adapt to downstream tasks in a low-dimensional subspace, which allows to parameterize the update by low-rank factors. Low-rank finetuning requires minimal trainable parameters and significant reduction of memory and computation resources [10, 12]. A number of works [14, 53, 18, 29, 30, 34, 3] have emerged to further improve the efficiency and adaptation capacity of LoRA.

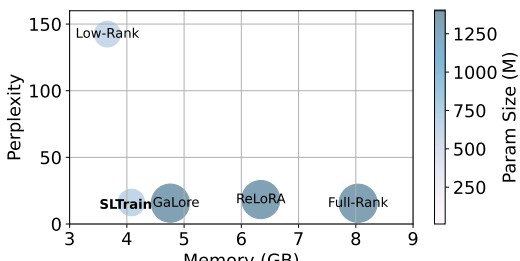

Figure 1: Shown are perplexity, memory, and parameter size for pretraining LLaMA 1B on the C4 dataset with different methdods. The radius and color of each circle scale with parameter size. Overall, the methods which have smaller, lighter circles on the left bottom corner are desirable for pretraining. The details are in Section 5.1.

While most of the works have focused on exploiting low-rank structure for fine-tuning, only a few [27, 24, 43, 47] have considered pretraining with low-rank weights. It has been observed that the performance of low-rank training often lags behind full-rank training despite the great potential for improving training and memory efficiency [47, 59]. This is because neural networks often exhibit full-rank structure in the weights and imposing low-rank restrictions could significantly limit their representation power. Hence, recent works have explored full-rank training with low-rank updates. For instance, ReLoRA [32] periodically restarts LoRA, where the low-rank updates are merged with the weights from the last period. However, ReLoRA also requires a warm-start full-rank training to achieve competitive performance [32]. GaLore [59] takes a different route by enforcing a low-rank structure not on the weights but on the gradients. This allows the Adam optimizer states to be stored in a low-dimensional space. While being memory efficient, GaLore is not parameter efficient because it still performs full parameter update with "projected-back" low-rank gradients.

*Parameter efficiency* is a desirable property post-pretraining for model deployment, fine-tuning, and model storage. On the other hand, *memory efficiency* is necessary for training models with lower hardware requirements. Despite the importance of both parameter and memory efficiency, these two goals are often pursued independently. While low-rank models achieve both parameter and memory efficiency, as discussed earlier, they do not perform well in general [47, 59]. Therefore, a natural question is:

*how can we adapt low-rank training to achieve comparable performance as full-rank training while maintaining both parameter and memory efficiency?*

**Contributions.** In this work, we answer the above question by directly parameterizing the weights as low-rank plus sparse factors for pretraining. It should be noted that both low-rank and sparse factors individually facilitate parameter efficiency. Furthermore, their combination (usually) ensures that the final pretrained model is of high rank. Existing strategies for sparse learning usually involve prune-and-growth [13, 2, 58, 49] that iteratively train, prune, and grow neurons. Such a strategy is usually not memory efficient due to the need of storing (and learning) a support and a dense weight matrix. In contrast, we motivate and adopt a simpler strategy of fixing a uniformly random support (for the sparse factor). This allows to only store indices and values for memory efficient training, which scales with the number of nonzero entries. We show such a simple approach allows to further reduce the memory consumption during pretraining compared to ReLoRA [32] and GaLore [59] without sacrificing performance. We show this through an extensive set of experiments on the LLaMA language models with varying model size from 60M up to 7B parameters. We call our proposed **s**parse plus **l**ow-rank pre**train**ing algorithm as **SLTrain**. In Figure 1, we observe that SLTrain obtains perplexity score comparable to full-rank model with considerable memory and parameter efficiency.

We end this section by noting that the idea of marrying low-rank and sparse factors has been explored for robust matrix recovery [6, 57, 4], attention matrix approximation [7], and neural network compression [31]. However, it is introduced for pretraining LLMs for the *first* time in our work.

## 2 Background on low-rank pretraining

Existing pretraining works [24, 43] have explored low-rank parameterization of the layer weights directly as $W = BA$. However, it has been empirically observed that vanilla low-rank parameterization suffers from large performance degradation because of the limited representation capacity [47, 32, 59]. Hence, motivated from low-rank adaptation (LoRA) [21] for fine-tuning, for pretraining, ReLoRA [32] suggests to parameterize the layer weights as

$$W = W_0 + \sum_{s=1}^{m} B_s A_s, \tag{1}$$

where $m$ represents the number of low-rank factors. This parameterization results in an overall high-rank update compared to LoRA because the sum of low-rank matrices is generally a higher rank matrix. The optimization is performed by training $B_s, A_s$ iteratively, merging $W_s \leftarrow W_{s-1} + B_s A_s$, and then restarting the optimization for $B_{s+1}, A_{s+1}$. A key drawback of ReLoRA is that it stores the full-rank matrix $W_s$ throughout the training and inference stages. Hence, it is memory intensive and not parameter efficient. While ReLoRA performs sequential low-rank updates in (1), a recent work [22] has explored parallel low-rank updates and merging them for pretraining.

A more recent work, GaLore [59], imposes low-rank structure on the gradient. Specifically, GaLore still optimizes full-rank weights and computes full-rank gradients $G_t$ at iteration $t$, but updates Adam moments $M_t, V_t$ in a low-dimensional space, i.e.,

$$M_t \leftarrow \beta_1 M_{t-1} + (1 - \beta_1) P_t^\top G_t, \quad V_t \leftarrow \beta_2 V_{t-1} + (1 - \beta_2)(P_t^\top G_t)^2$$
$$W_{t+1} \leftarrow W_t - \eta\, P_t M_t / (\sqrt{V_t} + \epsilon),$$

where $P_t$ is a projection matrix constructed by taking the largest left singular vectors of $G_t$. To reduce computational cost, $P_t$ is computed every several iterations and is stored in the middle. Although being memory efficient (as $M_t$ and $V_t$ are computed in the smaller dimension), GaLore is not parameter efficient due to computation of $P_t M_t$ for updating $W_t$.

## 3 SLTrain: proposed sparse plus low-rank pretraining

In order to achieve both parameter and memory efficiency, we propose to adapt low-rank parameterization by introducing a sparse factor. We model the weight matrices as a sum of sparse and low-rank matrices. Our proposed modeling is referred to as SLTrain. Below, we discuss the motivation, modeling details, and practical considerations for implementing SLTrain.

### 3.1 Motivation for sparse plus low-rank parameterization

Both low-rank and sparsity are parsimonious modeling strategies for exploring low-dimensional weight matrices. The low-rank component aims to learn the low-dimensional bases or eigenspaces of the weights. The sparse component, on the other hand, identifies effective neuron-wise interactions and disregards non-expressive ones. In linear algebra terms, the low-rank component enforces sparsity of singular values, whereas the sparse component enforces sparsity of individual entries. In general, low-rank matrices are not sparse, and sparse matrices are not necessarily low-rank [6]. These concepts provide complementary information that should be explored simultaneously.

Despite that low-rank modeling alone can have limited expressivity due to the low-rank structure it imposes, we show in the below proposition that low-rank plus a uniform sparse matrix with only $\Omega(\log n/n)$ number of entries is full-rank with high probability.

**Proposition 1.** *Consider a matrix $S \in \mathbb{R}^{n \times n}$ with support $\mathcal{S}$ sampled uniformly at random with probability $\delta \in (0, 1)$, i.e., $\mathbb{P}[(i, j) \in \mathcal{S}] = \delta$, for all $i, j \in [n]$. Suppose $\delta = \Omega(\log n/n)$, then with probability at least $1 - O(1/n)$, $BA + S$ is full rank for arbitrary randomly generated $B, A$.*

To further motivate the sparse plus low-rank modeling, in Figure 2, we illustrate different statistics from weight matrices of a pretrained LLaMA 60M model on C4 dataset (introduced later in Section 5). In Figure 2(a), we plot the singular values of weight matrices of different layers. The plot exhibits a fast decay of the singular values followed by a more stable decay of (smaller) singular values. This suggests that the top subspaces can be effective in model compression, and therefore, builds a case

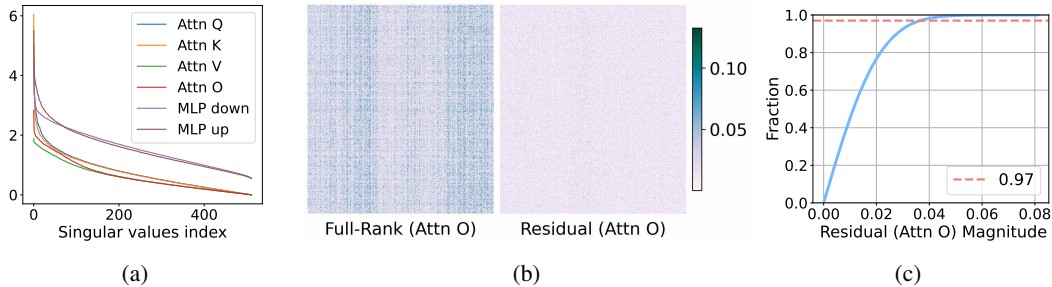

(a)                 (b)                 (c)

Figure 2: Illustration of the last attention layer of pretrained full-rank LLaMA 60M model on 1.1B tokens. (a): singular value magnitudes of weight matrices where we observe a rapid decay of singular values. (b): Visualization of full-rank pretrained attention output matrix $W_0$ in magnitude and the residual matrix after removing the best rank-$r$ ($r = 128$) approximation of the $W_0$ by SVD. We observe the magnitudes of the residual vary smoothly across different neuron-neuron interactions. (c): Cumulative density of the residual matrix in magnitude where we include a cut-off fraction at 0.97. We observe 97% entries in the residual matrix have magnitude less than 0.04.

for low-rank modeling. However, the tail singular value distribution shows that low-rank modeling purely may not be sufficient. In order to better understand the tail part, in Figure 2(b), we visualize the magnitude of the attention output weight matrix before and after we extract the top $r$-dimensional subspaces ($r = 128$) for the last attention layer. It is apparent that, after removing the top subspaces, both the magnitudes and the variation of the entries present in the residual matrix become smaller. Plotting the magnitudes of the entries in Figure 2(c) we see that 97% of the entries have a magnitude below 0.04. In Appendix B and C, we provide such visualizations for other layers of LLaMA 60M and Pythia 70M to further corroborate the findings. Overall, the figures suggest that a sparse matrix with random support can approximate the residual well given the magnitudes do not vary too much across the entries.

In Table 1, we perform an ablation study that verifies the feasibility of using a random sparse support for approximating the residual matrix. Specifically, we take $L_0$ as the best rank-$r$ approximation ($r = 128$) for the pretrained weight matrix $W_0$ and evaluate the perplexity score (PPL) on the validation set. We see that compared to the full-rank pre-trained model, low-rank approximation $L_0$ suffers from a drastic performance drop. We also augment the low-rank approximation $L_0$ with either top 3% or random 3% of entries of the residual matrix, which we label as top sparse or random sparse pruning, respectively.

We observe that $L_0$ plus top sparse pruning performs better compared to $L_0$ plus random sparse pruning. Nonetheless, both the performance is poor. We further evaluate fixing the low-rank approximation (to $L_0$) and only optimizing the sparse components with either top support or random support (both run for five times). Averaged PPL (over the five runs) for both the approaches improve and are comparable. This shows that fixing a random support for the sparse factor is a promising strategy from both efficiency and performance point of view. We explore learning both the sparse and low-rank factors in the next section.

Table 1: Perplexity (PPL) of training and pruning with random versus top sparsity for LLaMA 60M on 1.1B tokens.

|  | PPL ($\downarrow$) |
| --- | --- |
| Full-rank | 34.06 |
| Low-rank ($L_0$) | 36633.04 |
| $L_0$ + top sparse pruning | 5293.93 |
| $L_0$ + random sparse pruning | 29121.38 |
| $L_0$ + sparse training with top support | 53.75 |
| $L_0$ + sparse training with random support | 51.98 |

## 3.2   Our proposed modeling

Building on Section 3.1, we propose to parameterize weight matrices $W \in \mathbb{R}^{d \times p}$ as

$$W = BA + S,$$

where $B \in \mathbb{R}^{d \times r}, A \in \mathbb{R}^{r \times p}$ are low-rank factors with $r < \min\{d, p\}$ being the rank parameter and $S \in \mathbb{R}^{m \times n}$ is a sparse matrix. The number of non-zero entries (nnz) in $S$ is determined by the sparsity level parameter $\delta \in (0, 1)$, i.e., $\mathrm{nnz}(S) = \delta dp$. So, the total number of parameters for the proposed parameterization is $(d + p)r + \delta dp$, which is much smaller than the full-rank layer parameters $dp$ when we choose $\delta \ll 1$. In addition to being parameter efficient, the optimization states also cost less memory and scales with the number of trainable parameters. Finally, we note that the overall rank of $W$ will generally be high due to the presence of the sparse factor $S$, based on Proposition 1.

The performance of such a parameterization highly depends on whether there exists an implementation that is both computation and memory efficient. Nevertheless, modern GPU hardware is not suited for sparse tensor multiplication $Sx$ for given input $x$, as well as its gradient, especially when $S$ presents an unstructured sparsity pattern [7]. This causes increased computational bottleneck despite showing memory advantage. Thus, existing works on sparse network and training mostly rely on learning and storing a parameter mask (i.e., support) [48, 15, 33] by letting $S = M \odot U$, where $M \in \{0, 1\}^{d \times p}$ is a binary mask and $U \in \mathbb{R}^{d \times p}$ is a dense parameter. This allows to exploit GPU accelerator for dense matrix computation. However, masking requires to store both the support and a dense parameter for training, which significantly increases the memory cost.

In this work, we achieve memory efficiency by representing $S$ in terms of its indices and values, i.e., $(\mathcal{I}, \mathcal{V}) \in \mathbb{R}^{2\mathrm{nnz}(S)}$. This is possible because we randomly (and uniformly) fix the support a priori. The motivation for using a random (but fixed) support comes from the usefulness of random support in Table 1. This ensures the memory scales with only the sparsity in $S$ (i.e., the support size) rather than the full size of $S$. Further, the forward pass involves computing

$$BAx + Sx = (BA \oplus_{\mathcal{I}} \mathcal{V})x,$$

where we denote $W \oplus_{\mathcal{I}} \mathcal{V}$ as scatter-adding $\mathcal{V}$ to $W$ at the indices specified in $\mathcal{I}$. Because this operation results in a dense matrix, sparse matrix multiplication is avoided. Hence, this is GPU friendly without requiring to store a binary mask.

We remark that despite we require to compute a dense matrix, which has the same size as the full-rank matrix, we *never* store it for backpropagation. In particular, we can compute the gradient with respect to $B, A, \mathcal{V}$, and input $x$ as

$$\nabla_B L = \nabla_z L\, x^\top A^\top, \ \nabla_A L = B^\top \nabla_z L\, x^\top, \ \nabla_{\mathcal{V}} L = (\nabla_z L\, x^\top)_{\mathcal{I}}, \ \nabla_x L = (BA \oplus_{\mathcal{I}} \mathcal{V})^\top \nabla_z L, \tag{2}$$

where we let $z = (BA \oplus_{\mathcal{I}} \mathcal{V})x$ and $L$ denotes the loss function. We also denote $W_{\mathcal{I}}$ as gathering the values of $W$ at indices $\mathcal{I}$. In other words, we only need to store $B, A, \mathcal{I}, \mathcal{V}$ for backpropagation. This is illustrated in Algorithm 1 where we define a customized linear layer in SLTrain. We highlight that such a parameterization is agnostic to the chosen optimizers and can easily be integrated with any optimizer including Adam.

In comparison with the recent pretraining works based on low-rank factors/gradients, SLTrain is more parameter and memory efficient than ReLoRA [32] and GaLore [59] as it only optimizes the low-rank and sparse factors without the need for storing full-rank matrices.

### 3.3 Practical considerations

**Initialization and scaling.** We consider LoRA type of initialization for low-rank factors, i.e., Kaiming initialization [19] for $A$ factor and zero initialization for $B$ factor. For sparse factor, we adopt uniform initialization for the values $\mathcal{V}$ in the range of $[-1/\sqrt{d_{\mathrm{in}}}, 1/\sqrt{d_{\mathrm{in}}}]$, where $d_{\mathrm{in}}$ denotes input feature dimension. We choose the sparse support $\mathcal{I}$ uniformly at random up to the desired sparsity level $\delta$. Furthermore, in order to balance the contribution of low-rank factor and sparse factor, we follow LoRA [21] to scale the low-rank factors by $\alpha/r$ where the balancing parameter $\alpha$ is a hyperparameter. This hyperparameter, along with the stepsize has a joint effect on the training speed of low-rank versus sparse factors.

---

**Algorithm 1** SLTrain for linear layer

---
1: **Input:** $x, B_t, A_t, (\mathcal{I}, \mathcal{V}_t)$.
2: **def** forward($x, B_t, A_t, \mathcal{I}, \mathcal{V}_t$):
3:     save_for_backward($B_t, A_t, \mathcal{I}, \mathcal{V}_t$)
4:     **return** $(BA \oplus_{\mathcal{I}} \mathcal{V})x$
5:
6: **def** backward($\nabla_z L$):
7:     $x, B_t, A_t, \mathcal{I}, \mathcal{V}_t \leftarrow$ saved_tensor
8:     Compute gradient as in (2).
9:     **return** $\nabla_x L, \nabla_{B_t} L, \nabla_{A_t} L, \text{None}, \nabla_{\mathcal{V}_t} L$

---

**Regularization and preconditioning.** It is expected that the optimization of low-rank factors can cause instability when using larger stepsize or larger balancing parameter $\alpha$, an issue already present in low-rank training [43]. This is primarily due to the multiplicative updates of $B, A$ simultaneously. Existing solutions, such as orthogonal constraints or regularization [43], preconditioning [50, 23, 56], can be easily combined with the proposed modelling for more stable convergence.

**Integration with other techniques.** Since the proposed sparse plus low-rank approach pursues memory saving from the perspective of reparameterization, SLTrain can be easily integrated with optimizer-based techniques for further improving memory efficiency, including quantization [9] (that uses lower-bits for storing moment states without sacrificing the performance), per-layer weight updates [36] (that updates the parameters along with backpropagation), and activation checkpointing (that recomputes the activation states instead of storing them). In addition, SLTrain can be even combined with low-rank gradients in GaLore [59] for low-rank factors. This can further reduce the memory footprint, especially for larger models where the rank $r$ is set to be high. On the other hand, because we use a simple strategy of fixed-support sparse learning, it may be beneficial to combine with different techniques for dynamic support learning [2, 20].

## 4 Related works

**Low-rank fine-tuning and training.** Building on the idea of LoRA [21] that parameterizes the update as low-rank factors, i.e., $\Delta W = BA$, ROSA [14] dynamically adapts subspaces for training, where the subspaces are selected by taking SVD of the current weight matrices. Chain of Lora [53] decomposes the low-rank update into a sequence of small-size matrix product, $\Delta W = \sum_{j=1}^{k} B_j A_j$. NOLA [29] parameterizes the two small matrices as linear combination of two sets of random basis respectively, $B = \sum_{i=1}^{m} \alpha_i B_i, A = \sum_{j=1}^{n} \beta_j A_j$ where $A_i, B_j$ are fixed random matrices. NOLA optimizes over the coefficients, thus further improving parameter efficiency. VeRA [30] considers a similar parameterization as NOLA where $B = \mathrm{diag}(b)\widetilde{B}, A = \mathrm{diag}(a)\widetilde{A}$ for fixed random matrices $\widetilde{B}, \widetilde{A}$. DoRA [34] decomposes pre-trained weights into magnitude and directional components and separately fine-tune with LoRA adopted for directional update. SoRA [11] introduces a dynamic rank adaptation strategy for tuning LoRA rank. ResLoRA [46] adds a residual path for LoRA adaptors. For *pretraining*, in addition to ReLoRA [32] and GaLore [59], Flora [17] demonstrates LoRA updates approximate random projection of gradient, and by resampling the random projection, high-rank training can be achieved. LTE [22] adopts the similar idea of parameterizing a high-rank matrix through summation of low-rank matrices and adopts the parallel train-and-merge strategy as opposed to sequential in ReLoRA [32].

**Sparse fine-tuning, training and sparse networks.** Sparse fine-tuning/training aims to selectively update the weights with others fixed [48, 1, 2, 49, 15, 33]. This usually entails choosing a proper subset of parameters either randomly [49], or based on approximate Fisher information [1], magnitude of the change [48], gradients and momenta [2], as well as by learning a parameter mask (for storing support) with sparsity regularization [15]. On the other hand, sparse networks, also known as model pruning, directly search for a minimal architecture [16, 35] by removing redundant weights. We refer to the survey [20] for complete discussions of sparse network pruning.

**Sparse plus low-rank.** Decomposing a matrix into the sum of low-rank and sparse matrix is a classic optimization problem for matrix recovery [6, 54, 4]. Recently, some works also consider harnessing both low-rank structure and sparsity for neural network compression. Scatterbrain [7] considers approximating the attention matrix for faster inference with sparse plus low-rank factors. More specifically, given $Q, K, V \in \mathbb{R}^{n \times d}$ The main aim is to efficiently approximate $\exp(QK^\top)V$, which suffers from quadratic complexity in sequence length $n$. Hence, [7] proposes to leverage a random feature map $\phi : \mathbb{R}^d \to \mathbb{R}^m$, defined as $\phi(x) = \frac{1}{\sqrt{m}} \exp(Wx - \|x\|^2/2)$ with entries of $W$ sampled from Gaussian distribution $\mathcal{N}(0, 1)$, which defines a low-rank approximation $\phi(Q)\phi(K)^\top \approx \exp(QK^\top)$. Then a sparse matrix is constructed based on locality sensitivity hashing with the non-zero entries $S_{i,j} = \exp(QK^\top)_{i,j} - \phi(Q)_i^\top \phi(K)_j$. However, the aim of [7] is to approximate the attention matrix to reduce the computational cost while we aim to achieve memory efficiency by directly parameterizing the weight matrix. More specifically, in the context of self-attention where $Q = XW_Q, K = XW_K, V = XW_V$, we directly parameterize each projection matrix $W_Q, W_K, W_V$ as low-rank plus sparse factors, e.g., $W_Q = BA + S$. In addition, LoSparse [31] proposes to decompose the pretrained weights into low-rank plus sparse factors for structured

Table 2: Validation perplexity (PPL($\downarrow$)), number of parameters in millions (Param), and estimated total memory cost in G (Mem). The perplexity results for all the baselines are taken from [59]. For SLTrain, we use the same rank as other baselines and fix $\delta = 0.03$.

| | 60M | | | 130M | | | 350M | | | 1B | | |
|---|---|---|---|---|---|---|---|---|---|---|---|---|
| $r$ / $d$ | 128 / 512 | | | 256 / 768 | | | 256 / 1024 | | | 512 / 2048 | | |
| Tokens | 1.1B | | | 2.2B | | | 6.4B | | | 13.1B | | |
| | PPL | Param | Mem | PPL | Param | Mem | PPL | Param | Mem | PPL | Param | Mem |
| Full-Rank | 34.06 | 58 | 0.35 | 24.36 | 134 | 0.81 | 18.80 | 368 | 2.21 | 15.56 | 1339 | 8.04 |
| Low-Rank [24] | 78.18 | 43 | 0.24 | 45.51 | 94 | 0.57 | 37.41 | 185 | 1.11 | 142.5 | 609 | 3.66 |
| ReLoRA [32] | 37.04 | 58 | 0.36 | 29.37 | 134 | 0.84 | 29.08 | 368 | 1.85 | 18.33 | 1339 | 6.34 |
| GaLore [59] | 34.88 | 58 | 0.28 | 25.36 | 134 | 0.61 | 18.95 | 368 | 1.59 | 15.64 | 1339 | 4.76 |
| SLTrain | 34.15 | 44 | 0.26 | 26.04 | 97 | 0.60 | 19.42 | 194 | 1.24 | 16.14 | 646 | 4.16 |

compression. Nevertheless, they consider optimizing the sparse matrix via iterative thresholding, which requires to store the full-size sparse matrix. We instead consider directly optimizing the sparse matrix on its non-zero entries for memory-efficient pretraining.

**Memory efficient training.** To overcome the memory limitation of LLMs, many techniques have been proposed, such as reduced-precision, quantization [9, 10], gradient checkpointing [42] and gradient accumulation [8], and row-sum/column-sum of second-order statistics in Adafactor [45], among many others. As has already been discussed, the proposed sparse plus low-rank parameterization is orthogonal to these developments where the techniques can be easily integrated for further memory reduction.

## 5 Experiments

This section validates the effectiveness of low-rank plus sparse structure for pretraining and fine-tuning large language models. All the experiments are run on NVIDIA A100 GPUs. The code is available on `https://github.com/andyjm3/SLTrain`.

### 5.1 Pretraining LLMs

Following [32, 59], we consider pretraining the LLaMA language models [51] with pre-normalization, RMSnorm [55], and SwiGLU activation [44]. We train LLaMA LLMs on C4 (Colossal Clean Crawled Corpus) dataset [41], which is specially designed for pretraining. The training is performed without data repetition and we consider LLaMA with varying model sizes from 60M up to 7B parameters.

**Baselines.** We compare our SLTrain with the baselines which exploit low-rank structures.

- **Full-Rank**: This is the vanilla baseline that pretrains with full-rank weights using the Adam optimizer.
- **Low-Rank** [24]: This is the low-rank parameterization of weights by factorizing $W = BA$ where optimization is on $B, A$.
- **ReLoRA** [32]: ReLoRA periodically restarts LoRA [21] by merging the learned low-rank adaptors with layer weights and reinitializing the optimizer state and learning rate.
- **GaLore** [59]: GaLore explores low-rank structure for the gradients rather than for the parameters.

We implement SLTrain with Adam by reparameterizing the weights from all linear layers, including fully-connected layers as well as query, key, value projection layers. The remaining parameters are updated with full-rank parameterization. This is consistent with the setup used in [21, 32, 59].

**Hyperparameters.** For SLTrain, we fix the rank $r$ to be the same as the baselines and fix sparsity ratio $\delta = 0.03$ across all the model sizes except for LLaMA 7B where we choose $\delta = 0.05$, which achieves a good balance between efficiency and performance. We tune and fix the stepsize to be

0.003 and tune $\alpha$ in the range of $[8, 16, 32]$ for the LLaMA 60M, 130M, 250M, 1B models. We fix the other parameters to their default settings. In particular, we choose $\alpha = 32$ for the LLaMA 60M model and $\alpha = 16$ for the 130M and 350M models and $\alpha = 8$ for 1B. For the LLaMA 7B model, we choose stepsize to be 0.0005 and $\alpha = 8$. Except for the 7B model, we directly inherit the perplexity results from [59] and thus do not need to tune the hyperparameters from the baselines. We ensure the comparison is fair based on the training token number.

**Memory cost estimation.** We compare the proposed SLTrain with the low-rank baseline models in terms of estimated memory consumption. Following [59], we compute memory estimates with `bfloat16` format, where each floating point number occupies 2 bytes. We remark that SLTrain stores the indices with `int64` format, which occupies 8 bytes per digit. The memory cost for a training algorithm consists of the parameter memory and optimizer state memory. The parameter memory refers to the memory occupied by storing parameters, and the optimizer state memory refers to the memory required to store the first and second-order moment statistics, e.g., in Adam. Table 2 reports the total estimated memory cost for each method. The detailed breakdown of memory estimation can be found in Appendix F.

**Perplexity vs efficiency.** In Table 2, we compare the performance of different methods in three aspects: perplexity score, parameter size, and memory cost. We observe that SLTrain performs comparatively as the full-rank training and GaLore [59] while achieving further reduction in parameter size and memory cost. In addition, SLTrain only adds a small parameter and memory overhead to the low-rank parameterization, yet significantly improves the perplexity score. Hence, learning the additional sparse factor indeed helps in strengthening the representation capacity of SLTrain. This intuition is also validated in Figure 10 (Appendix D), where we plot the singular value distribution for different weight matrices. Due to the presence of the sparse factor, we observe that the spectrum of the SLTrain weight matrices gets enhanced beyond $r = 128$.

**Measuring actual memory footprint.** In Figure 3, we record the actual memory footprint of different methods across various model sizes, on a single A100 80G GPU. We measure the memory of 8-bit SLTrain with per-layer weight update using a single batch size and `bfloat16` data type. Gradient checkpointing is not used for any method. The baselines include Adam trained on full-rank weights, 8-bit Adam, and 8-bit GaLore with per-layer weight. From the figure, we see that SLTrain achieves memory reduction by 51%, 58%, 73% for 350M, 1B, 7B models, respectively. Notably, compared to state-of-the-art memory-efficient method GaLore [59], SLTrain reduces the memory requirement by 29%, 34%, 17% when pretraining 350M, 1B, 7B models, respectively.

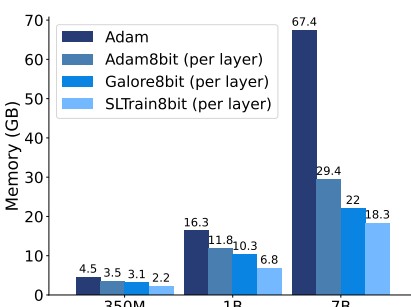

Figure 3: Actual memory consumption across different model size and algorithms on a single A100 80G GPU.

**Measuring throughput.** We measure the throughput of SLTrain for pretraining on the LLaMA 350M and LLaMA 1B models with a token batch size of 256 on 1×80G A100 GPU and 4×80G A100 GPUs, respectively. The throughput is averaged over 5000 training steps. We observe in Table 3 that the throughput token of SLTrain is slightly lower than the full-rank and GaLore baselines. This is mainly due to the retrieving and setting for the sparse entries. We believe more efficient implementation can be developed in this regard.

**Scaling to LLaMA 7B model.** For pretraining the LLaMA 7B model, due to the resource constraints, we only compare SLTrain with GaLore, implemented with 8-bit Adam [9] on 4×80G A100 GPUs, without per-layer weight updates nor gradient checkpointing.[1] We directly use the training scripts of GaLore.[2] In Table 4, we observe that SLTrain performs comparably to GaLore in terms of perplexity and throughput while achieving significant memory reduction of 26% per GPU device.

**Inference memory and throughput.** In Table 5, we compare the inference memory usage and throughput between SLTrain and the full-rank model across various model sizes, ranging from LLaMA 130M to 7B. A clear trade-off between memory and computational cost is observed. Specifically, as the model size increases, the percentage of memory savings becomes more pronounced, while

---

[1]For full-rank model, 8-bit Adam throws out-of-memory error on 4×80G A100 GPUs.
[2]The scripts are available at `https://github.com/jiaweizzhao/GaLore`

Table 3: Throughput tokens/seconds for LLaMA 350M (on 1×80G A100 GPU) 1B (on 4×80G A100 GPUs).

|  | 350M | 1B |
| --- | --- | --- |
| Full-Rank | 32072 | 20630 |
| GaLore | 31747 | 20500 |
| SLTrain | 30293 | 20412 |

Table 4: Validation perplexity, *actual* memory footprint per GPU, and throughput tokens/seconds (Tokens/sec) for LLaMA 7B on 1.4B tokens.

|  | PPL | Mem | Tokens/sec |
| --- | --- | --- | --- |
| 8-bit GaLore | 26.87 | 62G | 5924 |
| 8-bit SLTrain | 27.59 | 46G | 5877 |

Table 5: Inference memory and throughput comparison on a single 40G A100 GPU on a batch size of 128 for 130M, 350M, and a batch size of 32 for 1B, 7B. Compared to 1B model, higher memory for 350M model is due to the larger batch size.

|  | 130M | | 350M | |
|  | Mem | Tokens/s | Mem | Tokens/s |
| --- | --- | --- | --- | --- |
| Full-Rank | 8.09G | 151360 | 11.06G | 71324 |
| SLTrain | 7.95G | 137058 | 10.44G | 66616 |
|  | (-1.73%) | (-9.45%) | (-5.61%) | (-6.60%) |
|  | 1B | | 7B | |
|  | Mem | Tokens/s | Mem | Tokens/s |
| Full-Rank | 8.64G | 18964 | 32.93G | 9500 |
| SLTrain | 6.12G | 17482 | 21.19G | 8481 |
|  | (-29.17%) | (-7.81%) | (-35.65%) | (-10.73%) |

the corresponding increase in computational cost is less significant. This underscores the growing advantage of SLTrain when employing larger models.

**Varying random sparse support.** In contrast to common pruning strategies that require careful learning of sparse support, we show that SLTrain works with randomly selected support. In this regard, we perform pretraining on Llama 60M and 130M with five different randomly selected support for the sparse factor $S$. The convergence plots are shown in Figure 4, where we see that changing the random support for the sparse factor does not materially affect the performance.

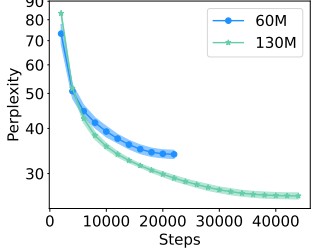

Figure 4: Convergence of SLTrain in perplexity with different random support.

**How do rank $r$ and sparsity $\delta$ affect performance?** In Table 6, we validate the performance of training the Llama 60M and 130M models by varying the hyperparameters $r$ and $\delta$. We notice that, in general, when more parameters are added (corresponding to higher $r$ and $\delta$), the performance is better. However, this performance gain is also accompanied by an increase in memory footprint. In our initial experiments, we also tried the extreme setting with $r = 0$, i.e., only the $S$ component was learned but with a higher $\delta$ value. This setting demonstrated a reasonably good performance. Further investigations in this direction are left for future work.

**Further comparisons to full-rank performance.** In this section, we evaluate the potential of SLTrain to achieve performance comparable to full-rank models by adjusting the sparsity ratio, $\delta$. As shown in Table 7, we increase $\delta$ from 0.03 to 0.1 for the LLaMA 350M and 1B models. Our results indicate that setting $\delta = 0.1$ enables SLTrain to attain perplexity scores similar to those of full-rank models, while maintaining memory and parameter efficiency. Notably, at $\delta = 0.1$, SLTrain reduces the parameter size by 42% for the 350M model and 45% for the 1B model. These results highlight the effectiveness of SLTrain in significantly reducing model size without sacrificing performance.

## 6   Concluding Remarks

In this paper, we propose SLTrain for achieving *both memory and parameter efficient* pretraining of LLMs. SLTrain combines two complementary parsimonious structures, low-rank and sparsity, for

Table 6: Ablation comparison with low-rank and sparse parameterization along with change of rank $r$ and sparsity $\delta$. Validation perplexity ($\downarrow$) and parameter size and estimated memory cost in brackets.

| | | **60M** | | **130M** |
|---|---|---|---|---|
| | | 1.1B | | 2.2B |
| Full-Rank | | 34.06 (0.35G) | | 24.36 (0.81G) |
| SLTrain | $r = 96, \delta = 0.03$ | 34.80 (0.25G) | $r = 224, \delta = 0.03$ | 26.25 (0.58G) |
| SLTrain | $r = 128, \delta = 0.01$ | 34.81 (0.26G) | $r = 256, \delta = 0.01$ | 26.50 (0.58G) |
| SLTrain | $r = 128, \delta = 0.03$ | 34.15 (0.26G) | $r = 256, \delta = 0.03$ | 26.04 (0.60G) |
| SLTrain | $r = 128, \delta = 0.05$ | 33.41 (0.28G) | $r = 256, \delta = 0.05$ | 25.72 (0.62G) |
| SLTrain | $r = 160, \delta = 0.03$ | 33.20 (0.28G) | $r = 288, \delta = 0.03$ | 25.93 (0.63G) |

Table 7: Results training LLaMA 350M (with batch size=128 per GPU) and LLaMA 1B (with batch size=32 per GPU). Validation perplexity (PPL) ($\downarrow$), number of parameters in millions (Param) and actual max memory allocated per GPU in G (Mem).

| | 350M | | | 1B | | |
|---|---|---|---|---|---|---|
| | PPL | Param | Mem | PPL | Param | Mem |
| Full-Rank | 18.80 | 368 | 59.34 | 15.56 | 1339 | 39.97 |
| SLTrain ($\delta = 0.03$) | 19.42 | 194 (-47%) | 57.90 (-2.4%) | 16.14 | 646 (-52%) | 33.77 (-15.5%) |
| SLTrain ($\delta = 0.05$) | 19.24 | 200 (-45%) | 58.00 (-2.2%) | 15.97 | 670 (-50%) | 34.30 (-14.2%) |
| SLTrain ($\delta = 0.1$) | 18.72 | 215 (-42%) | 58.25 (-1.8%) | 15.59 | 730 (-45%) | 35.36 (-11.5%) |

learning models with high representation capacity. While low rank is modeled via the product $BA$ of $r$-dimensional matrices, the support of the sparse factor $S$ is obtained by uniformly random sampling over indices. The matrices $B, A, S$ are learned for different layers via backpropagation. Empirically, we achieve state-of-the-art memory reduction during pretraining while maintaining competitive performance. Although we show results on the LLaMA language models (with varying size from 60M to 7B parameters), we believe SLTrain could also help to improve memory efficiency for vision foundation models and multi-modal foundation models, such as diffusion models [37] and CLIP [38]. The significant improvements shown by SLTrain also motivates future works to understand the theoretical guarantees of training with both low-rank and sparse factors, such as convergence and loss landscape. We hope this work initiates exploration on combination of other parsimonious structures for pretraining such as Kronecker product or structured sparsity (e.g., block-diagonal, group-lasso).

### Acknowledgments

M. Hong and J. Li are supported partially by NSF under the grants EPCN-2311007 and CCF-1910385, and an Amazon Research Award.

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

# Contents

## A  Proofs

*Proof of Proposition 1.*  For each row, the probability that at least one entry is non-zero is $1-(1-p)^n$ due to the independence of the non-zero entry. Due to the independence of uniform selection, the probability that all the rows have at least one entry is $(1-(1-p)^n)^n$. By choosing $p = 2\log n/n$, the probability can be simplified to $(1-e^{-2\log n})^n = 1 - O(1/n)$. Similarly, we can apply the same argument for the columns. Taking the union bound shows that with probability at least $1 - O(1/n)$, $S$ has at least one entry for each row and each column respectively. Further, because the non-zero entries of $S$ are sampled from continuous distribution, the set of matrices such that $S$ becomes low-rank has measure zero. Then taking union bound gives with probability at least $1 - O(1/n)$, $S$ is full rank.

The remaining part is to prove that $BA + S$ is full rank. Because $B$, $A$ are sampled from a continuous distribution space, the set of such matrices that augments $S$ to form a degenerate matrix has measure zero. Thus, the proof is complete. □

## B  Additional illustration for low-rank and residual factors

In the main text, we present singular values and visualizations of the attention output weight matrix for a single layer, both before and after applying the low-rank approximation. Here, we extend this analysis by providing visualizations for additional weight matrices across multiple layers for the LLaMA 60M and 130M models. The figures reveal a consistent pattern, showing a low-rank plus (uniformly) small magnitude structure in the weight matrices of other layers as well.

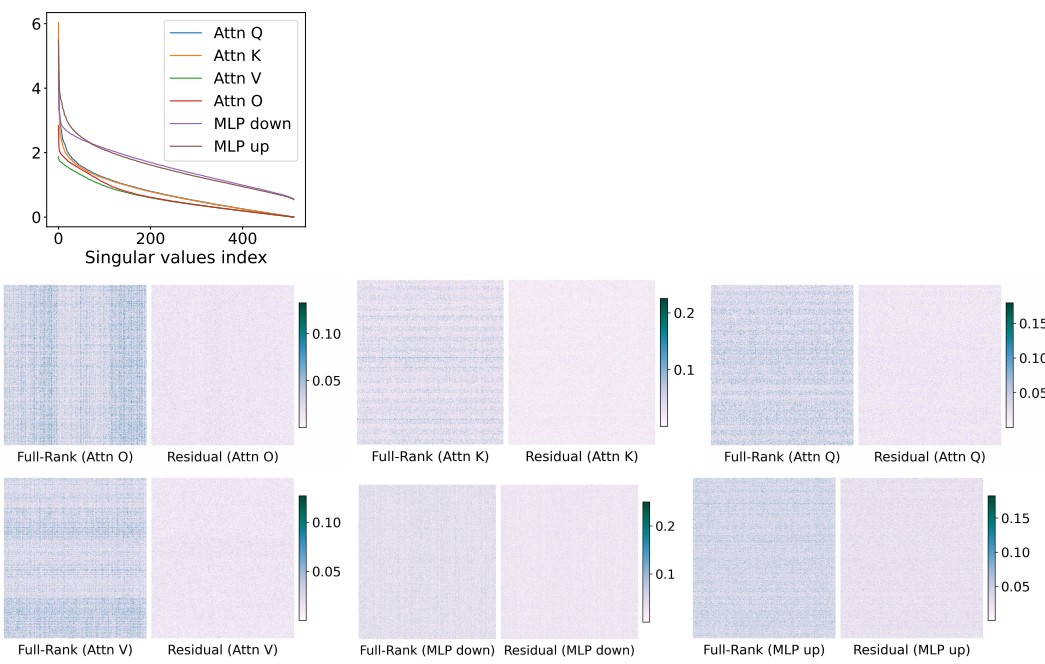

Figure 5: Illustration of the *last* attention layer of pretrained full-rank **LLaMA 60M** model on 1.1B tokens.

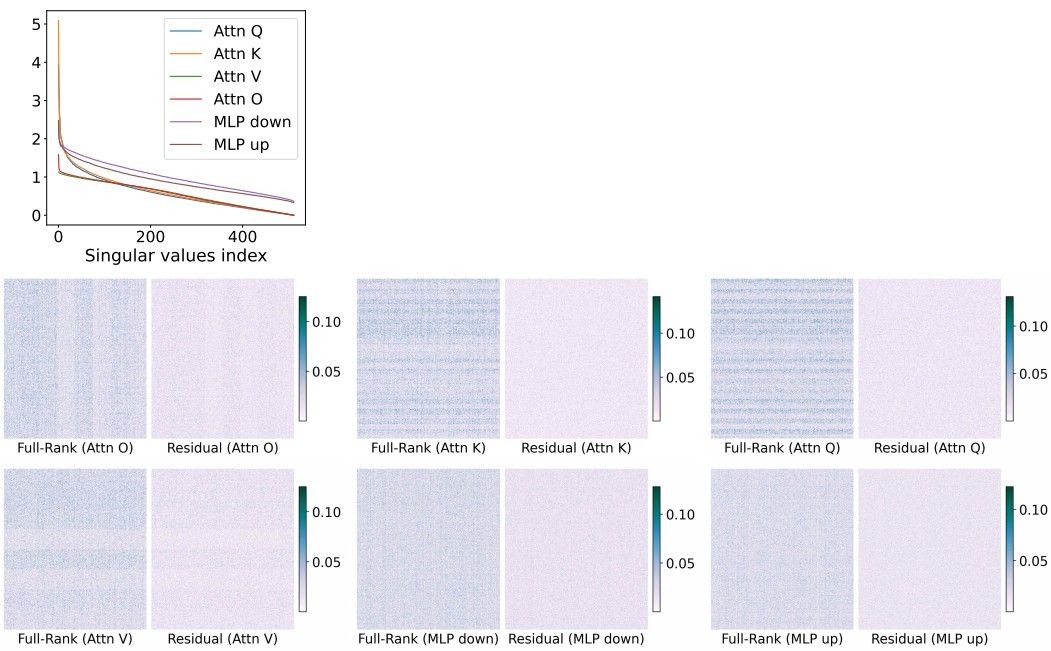

Figure 6: Illustration of the *first* attention layer of pretrained full-rank **LLaMA 60M** model on 1.1B tokens.

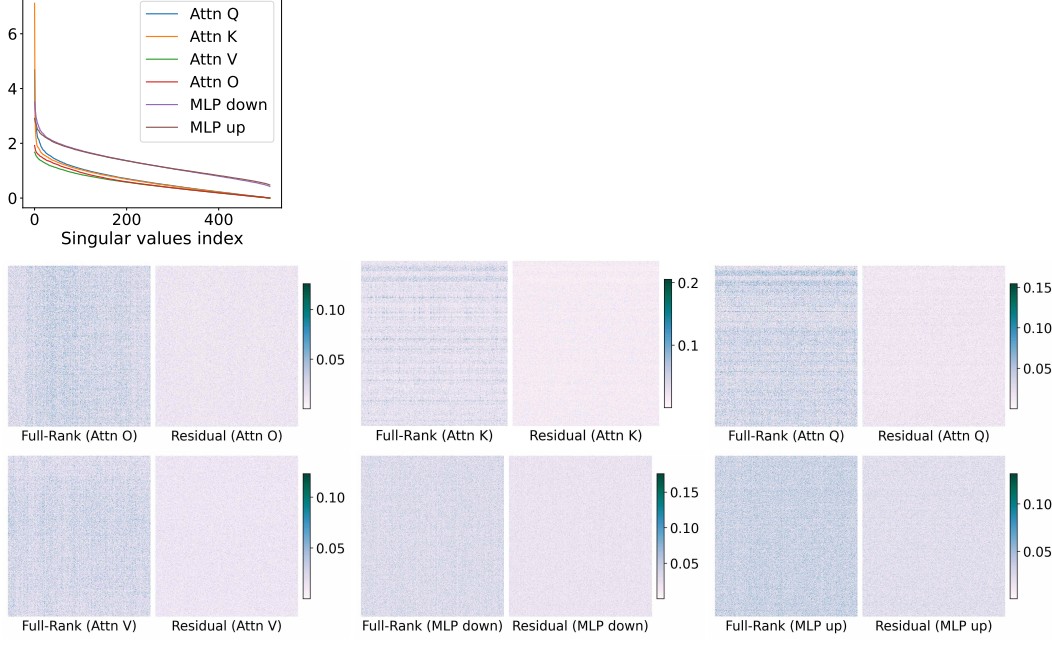

Figure 7: Illustration of the *fourth* attention layer of pretrained full-rank **LLaMA 60M** model on 1.1B tokens.

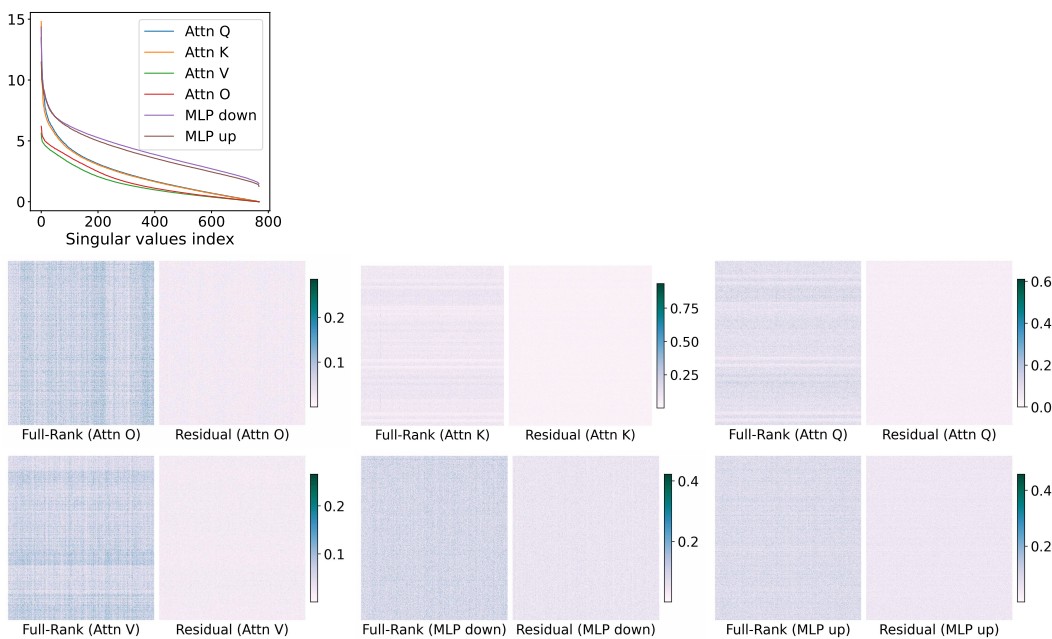

Figure 8: Illustration of the *eighth* attention layer of pretrained full-rank **LLaMA 130M** model on 2.2B tokens.

## C  Illustration of low-rank and residual factors for Pythia

Here we repeat the analysis of singular spectrum for pretrained Pythia 70M, downloaded from Hugging Face. Specifically, we set the rank $r = 128$ and extract the best rank $r$ approximation of the learned weight matrices. The results are shown in Figure 9, where we observe that the residual after removing the best low-rank approximation vary smoothly and has small magnitude.

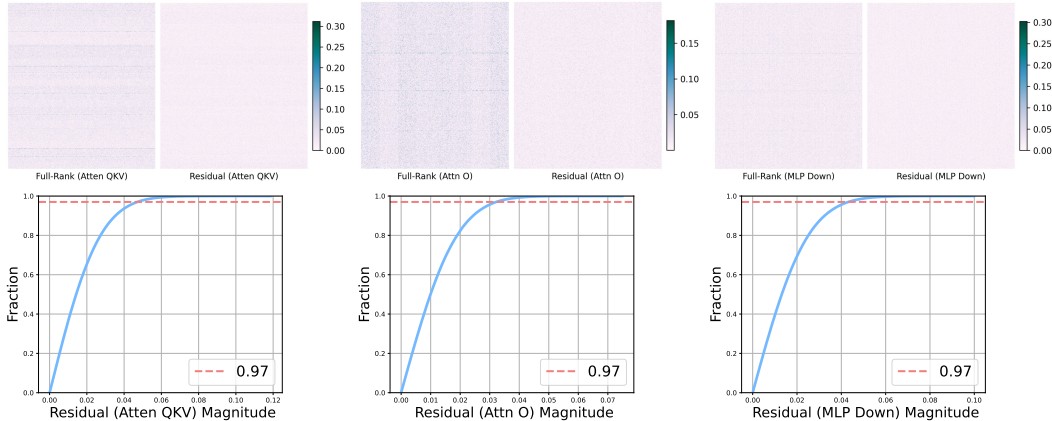

Figure 9: Illustration of the last layer of pretrained full-rank **Pythia 70M** (deduped) downloaded from Hugging Face.

# D    Singular value distribution of learned weights

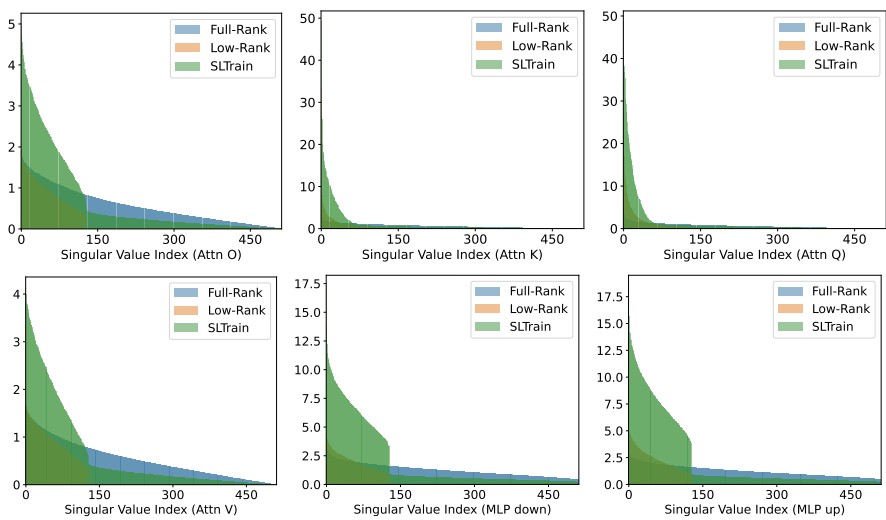

Figure 10: Visualization of singular value distribution for different weight matrices, pretrained on 1.1B tokens. SLTrain was trained with $r = 128$. For SLTrain, we see that the tail singular values (i.e., 129 till 512) is due to the sparse factor and the head singular values are because of the low-rank component. It is interesting to see that the tail distribution of SLTrain tries to follow that of Full-Rank. In the head distribution, SLTrain tries to approximate Low-Rank.

In Figure 10, we plot the distribution of singular values of pretrained weights of the LLaMA 60M model on 1.1B tokens. It is interesting to observe the distinct role of both the low-rank (L) and sparse (S) components. In particular, we see a cliff at index 128. This is because the singular values from 1 to 128 is due the learning of low-rank factor and from 129 to 512 is primarily due to the sparse component.

To further evaluate the contributions of spectrum from low-rank and sparse components, we decompose the singular values of SLTrain weight matrices into contributions from the low-rank and sparse parts. Specifically, let $U\Sigma V^\top = BA + S$ be the SVD, then we plot $\mathrm{diag}(\Sigma)$, $\mathrm{diag}(U^\top BAV)$, $\mathrm{diag}(U^\top SV)$, which we respectively call singular values, low-rank singular values, and sparse singular values. In Figure 11, we see that the low-rank and sparse components contribute to different ends of the spectrum of the weight matrices. This justifies our modeling approach, i.e., by adding a

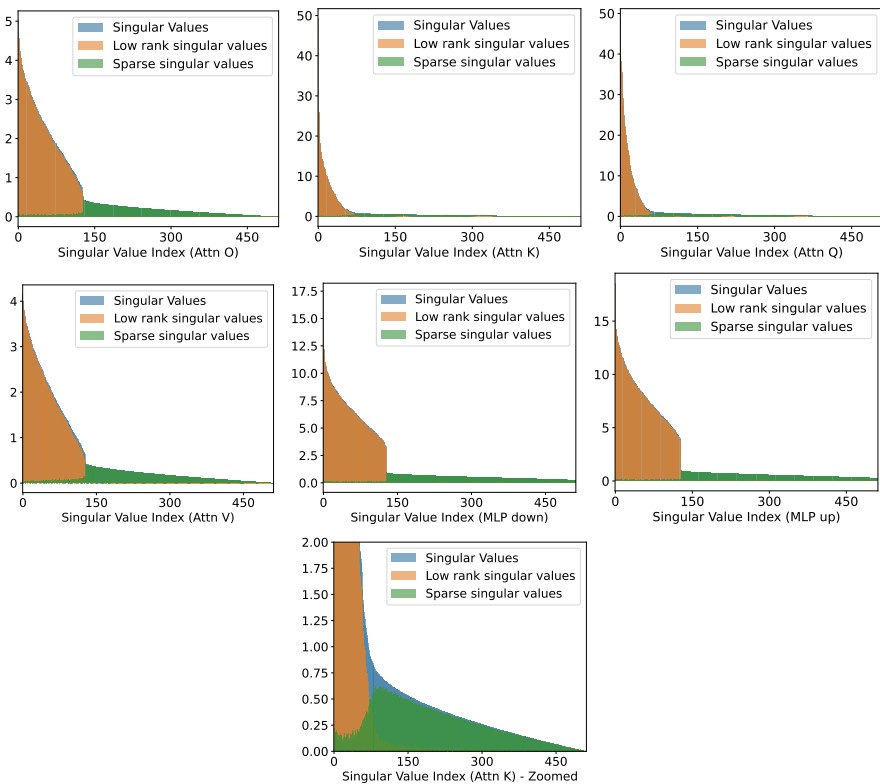

Figure 11: Visualization of singular value composition of SLTrain. In particular, we show the decomposition of singular values of learned $BA + S$ into contributions arising from from $BA$ and those from $S$. We clearly see that the top singular values are primarily due to the low-rank component and the tail singular values are due to the sparse component.

sparse component to the low-rank component, we mainly augment the tail end of the singular value spectrum.

## E  Memory and runtime of SLTrain linear layer

Here we perform an additional experiment by comparing the actual maximum memory consumption for the proposed SLTrain linear layer (Algorithm 1, $(BA + S)x$) and standard full-rank linear layer ($Wx$) and low-rank linear layer ($BAx$) in a feedforward neural network. We include the results for both forward and backward pass where we vary the number of layers in Figure 112. Specifically, we set the input, hidden and output size to be $2048$ and $r = 128$ with $\delta = 0.03$. From the figure, we observe that as the number of layers increases, the reduction in memory of SLTrain becomes more evident compared to full-rank model. On the other hand, the memory overhead of SLTrain compared to low-rank model is only marginal. In terms of computational cost, we see compared to full-rank model, SLTrain only requires slight computational overhead, which is due to the scatter-adding operation.

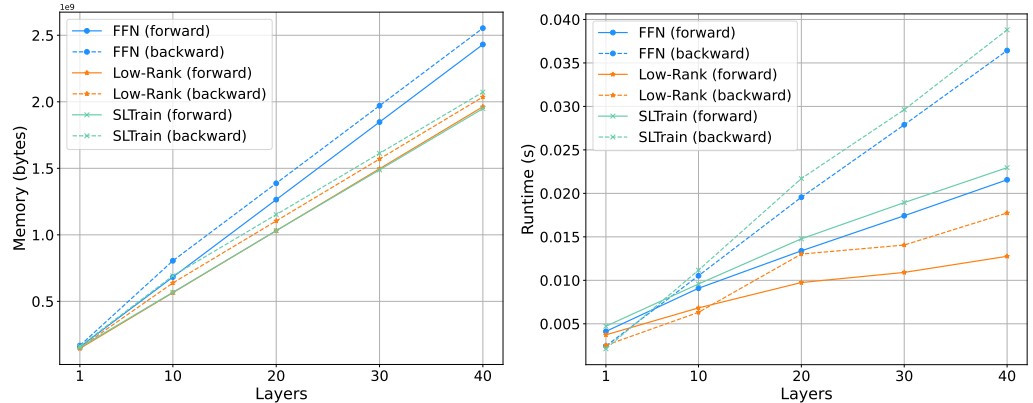

Figure 12: Comparison of memory and runtime with feedforward neural network with full-rank linear layer (FFN), low-rank linear layer ($BA$) and SLTrain linear layer ($BA + S$). We observe the memory savings of SLTrain becomes more evident when number of layers increases. Compared to Low-Rank, the memory of SLTrain is only marginally higher.

## F    Details of memory estimation

Following [59], we compute memory estimate with `bfloat16` format, where each floating point number occupies 2 bytes. For simplicity of estimation, we use the convention that 1G contains $10^9$ bytes. Hence the estimation could be different from the estimation in [59]. We summarize the parameter and optimizer state breakdown in Table 8.

For a **60M** model, we estimate the memory for each model as follows.

- **Full-rank**: Full rank model has 58.2M parameters, which costs 0.12G and the optimizer state requires to store double size of the parameters, which is 0.23G.

- **Low-rank**: A low-rank parameterization (with $r = 128$) for selected modules lead to 42.78M parameters (32.78M non-adapted parameters plus 10M low-rank adapted parameters), which consumes a memory of 0.08G. The optimizer state costs double size of memory, which is 0.16G.

- **ReLoRA**: Similar to LoRA, ReLoRA requires to store both the original parameters as well as the adaptors (including both the low-rank adaptors and adaptors for other parameters), which in total has 102.77M parameters with an estimated memory cost of 0.20G. For the optimizer state, ReLoRA stores the moments of only trainable parameters, which is 85.54M with a memory of 0.17G.

- **GaLore**: GaLore requires to store the full set of parameters 58.2M, which is the same as full-rank. For the optimizer state, GaLore stores the moments of projected gradients, which has a size of 78.20M, plus the projection matrix of size 3.67M. In total optimizer state requires to store 0.16G of memory.

- **SLTrain**: For proposed SLTrain (with $r = 128$ and $\delta = 0.03$), the parameter includes 32.78M base parameters, together with 10M low-rank factors and 0.76M sparse factors, which occupies 0.09G memory (where the sparse factors include 0.76M values in `bfloat16` format and 0.76M indices in `int64` format). For the optimizer state, the memory cost is double of the parameter size, which occupies 0.17G.

For a **130M** model, we estimate the memory for each model as follows.

- **Full-rank**: Full rank model has 134.11M parameters, which costs 0.27G and the optimizer state requires to store double size of the parameters, which is 0.54G.

- **Low-rank**: A low-rank parameterization (with $r = 256$) for selected modules lead to 94.00M parameters (49.17M non-adapted parameters plus 44.83M low-rank adapted parameters), which consumes a memory of 0.19G. The optimizer state costs double size of memory, which is 0.38G.

- **ReLoRA**: Similar to LoRA, ReLoRA requires to store both the original parameters as well as the adaptors (including both the low-rank adaptors and adaptors for other parameters), which in

Table 8: Breakdown of memory consumption of training modules in terms of parameters (Param) and optimizers (Optim).

| | 60M | | 130M | | 350M | | 1B | |
|---|---|---|---|---|---|---|---|---|
| | Param | Optim | Param | Optim | Param | Optim | Param | Optim |
| Full-rank | 0.12G | 0.23G | 0.27G | 0.54G | 0.74G | 1.47G | 2.68G | 5.36G |
| Low-rank | 0.08G | 0.16G | 0.19G | 0.38G | 0.37G | 0.74G | 1.22G | 2.44G |
| ReLoRA | 0.20G | 0.16G | 0.46G | 0.38G | 1.11G | 0.74G | 3.90G | 2.44G |
| GaLore | 0.12G | 0.16G | 0.27G | 0.34G | 0.74G | 0.85G | 2.68G | 2.08G |
| SLTrain | 0.09G | 0.17G | 0.21G | 0.39G | 0.46G | 0.78G | 1.58G | 2.58G |

total has 228.11M parameters with an estimated memory cost of 0.46G. For the optimizer state, ReLoRA stores the moments of only trainable parameters, which is 188M with a memory of 0.38G.

- **GaLore**: GaLore requires to store the full set of parameters 134.11M, which is the same as full-rank. For the optimizer state, GaLore stores the moments of projected gradients, which has a size of 154.97M, plus the projection matrix of size 16.52M. In total optimizer state requires to store 0.34G of memory.

- **SLTrain**: For proposed SLTrain (with $r = 256$ and $\delta = 0.03$), the parameter includes 49.17M base parameters, together with 44.83M low-rank factors and 2.55M sparse factors, which occupies 0.21G memory (where the sparse factors include 2.55M values in `bfloat16` format and 2.55M indices in `int64` format). For the optimizer state, the memory cost is double of the parameter size, which is 0.39G.

For a **350M** model, we estimate the memory for each model as follows.

- **Full-rank**: Full rank model has 367.97M parameters, which costs 0.74G and the optimizer state requires to store double size of the parameters, which is 1.47G.

- **Low-rank**: A low-rank parameterization (with $r = 256$) for selected modules lead to 185.22M parameters (65.59M non-adapted parameters plus 119.63M low-rank adapted parameters), which consumes a memory of 0.37G. The optimizer state costs double size of memory, which is 0.74G.

- **ReLoRA**: Similar to LoRA, ReLoRA requires to store both the original parameters as well as the adaptors (including both the low-rank adaptors and adaptors for other parameters), which in total has 553.19M parameters with an estimated memory cost of 1.11G. For the optimizer state, ReLoRA stores the moments of only trainable parameters, which is 370.44M with a memory of 0.74G.

- **GaLore**: GaLore requires to store the full set of parameters 367.97M, which is the same as full-rank. For the optimizer state, GaLore stores the moments of projected gradients, which has a size of 282.36M, plus the projection matrix of size 144.04M. In total optimizer state requires to store 0.34G of memory.

- **SLTrain**: For proposed SLTrain (with $r = 256$ and $\delta = 0.03$), the parameter includes 65.59M base parameters, together with 119.64M low-rank factors and 9.07M sparse factors, which occupies 0.46G memory (where the sparse factors include 9.07M values in `bfloat16` format and 9.07M indices in `int64` format). For the optimizer state, the memory cost is double of the parameter size, which is 0.78G.

For a **1B** model, we estimate the memory for each model as follows.

- **Full-rank**: Full rank model has 1339.08M parameters, which costs 2.68G and the optimizer state requires to store double size of the parameters, which is 5.36G.

- **Low-rank**: A low-rank parameterization (with $r = 512$) for selected modules lead to 609.31M parameters (131.17M non-adapted parameters plus 478.14M low-rank adapted parameters), which consumes a memory of 1.22G. The optimizer state costs double size of memory, which is 2.44G.

- **ReLoRA**: Similar to LoRA, ReLoRA requires to store both the original parameters as well as the adaptors (including both the low-rank adaptors and adaptors for other parameters), which in

total has 1948.39M parameters with an estimated memory cost of 3.90G. For the optimizer state, ReLoRA stores the moments of only trainable parameters, which is 1218.62M with a memory of 2.44G.

- **GaLore**: GaLore requires to store the full set of parameters 1339.08M, which is the same as full-rank. For the optimizer state, GaLore stores the moments of projected gradients, which has a size of 866.30M, plus the projection matrix of size 176.16M. In total optimizer state requires to store 2.08G of memory.

- **SLTrain**: For proposed SLTrain (with $r = 512$ and $\delta = 0.03$), the parameter includes 131.17M base parameters, together with 478.14M low-rank factors and 36.24M sparse factors, which occupies 1.58G memory (where the sparse factors include 36.24M values in `bfloat16` format and 36.24M indices in `int64` format). For the optimizer state, the memory cost is double of the parameter size, which is 2.58G.

In addition, we estimate the memory for Table 6 by listing out the parameter information for each parameter setting.

Table 9: Memory breakdown for SLTrain for LLaMA 60M with varying $r, \delta$.

|  | $r = 128, \delta = 0.01$ | $r = 128, \delta = 0.05$ | $r = 96, \delta = 0.03$ | $r = 160, \delta = 0.03$ |
|---|---|---|---|---|
| Total params | 43.02M | 44.04M | 41.03M | 46.03M |
| Base params | 32.78M | 32.78M | 32.78M | 32.78M |
| Low-rank parameters | 9.99M | 9.99M | 7.50M | 12.49M |
| Sparse parameters | 0.25M | 1.26M | 0.76M | 0.76M |
| Parameter memory | 0.09G | 0.10G | 0.09G | 0.10G |
| Optimizer memory | 0.17G | 0.18G | 0.16G | 0.18G |
| Total memory | 0.26G | 0.28G | 0.25G | 0.28G |

Table 10: Memory breakdown for SLTrain for LLaMA 130M with varying $r, \delta$.

|  | $r = 256, \delta = 0.01$ | $r = 256, \delta = 0.05$ | $r = 224, \delta = 0.03$ | $r = 288, \delta = 0.03$ |
|---|---|---|---|---|
| Total params | 94.85M | 98.24M | 90.94M | 102.15M |
| Base params | 49.17M | 49.17M | 49.17M | 49.17M |
| Low-rank parameters | 44.83M | 44.83M | 39.22M | 50.43M |
| Sparse parameters | 0.85M | 4.25M | 2.55M | 2.55M |
| Parameter memory | 0.20G | 0.23G | 0.20G | 0.22G |
| Optimizer memory | 0.38G | 0.39G | 0.36G | 0.41G |
| Total memory | 0.58G | 0.62G | 0.58G | 0.63G |

## G  Fine-tuning LLMs

In addition to pretraining, the sparse plus low-rank factor may also be used for fine-tuning LLMs as: $W = W_0 + BA + S$, where $W_0$ is a given pretrained model weight, $B$ and $A$ are low-rank factors, and $S$ is the sparse factor. As in Section 3.2, we learn the fine-tuned weights $B, A$, and $S$ and term this as SLTrain-FT. The fine-tuning experiments are conducted for RoBERTa base model (with 125M parameters) on GLUE benchmarks. We use $r = 8$ for all methods and tune the hyperparameters of SLTrain-FT as follows. We tune three hyperparmaters of SLTrain, i.e., sparsity level $\delta$, balancing parameter $\alpha$ and stepsize $\eta$. Specifically, we tune $\delta$ in the range of $[0.005, 0.001]$ and $\alpha$ in $[32, 64, 128]$, $\eta$ in [1e-5, 2e-5, 3e-5, 4e-5, 5e-5]. The tuned hyperparameters are in Table 11.

The results on benchmark datasets are shown in Table 12. We observe that SLTrain-FT performs competitively as the baselines. While there is no specific memory advantage of SLTrain-FT when a general full-rank $W_0$ is given, memory reductions can be obtained when $W_0$ is learned via SLTrain.

Table 11: Hyperparameters of SLTrain for fine-tuning. The batch size, number of epochs and rank $r$ follows from the choice in [59].

|  | CoLA | STS-B | MRPC | RTE | SST-2 | MNLI | QNLI | QQP |
|---|---|---|---|---|---|---|---|---|
| Batch Size | 32 | 16 | 16 | 16 | 16 | 16 | 16 | 16 |
| Epochs | 30 | 30 | 30 | 30 | 30 | 30 | 30 | 30 |
| Rank $r$ | 8 | 8 | 8 | 8 | 8 | 8 | 8 | 8 |
| Learning Rate | 3e-5 | 3e-5 | 5e-5 | 4e-5 | 1e-5 | 1e-5 | 3e-5 | 3e-5 |
| Sparsity $\delta$ | 0.005 | 0.005 | 0.001 | 0.001 | 0.005 | 0.001 | 0.005 | 0.005 |
| $\alpha$ | 32 | 64 | 32 | 128 | 32 | 128 | 128 | 32 |

Table 12: Results on GLUE benchmarks. Reported numbers for the baselines are directly retrieved from [59].

|  | CoLA | STS-B | MRPC | RTE | SST-2 | MNLI | QNLI | QQP | Avg |
|---|---|---|---|---|---|---|---|---|---|
| Full-rank FT | 62.24 | 90.92 | 91.30 | 79.42 | 94.57 | 87.18 | 92.33 | 92.28 | 86.28 |
| LoRA (rank=8) | 61.83 | 90.80 | 91.90 | 79.06 | 93.46 | 86.94 | 92.25 | 91.22 | 85.93 |
| GaLore FT (rank=8) | 60.06 | 90.82 | 92.01 | 79.78 | 94.38 | 87.17 | 92.20 | 91.11 | 85.94 |
| SLTrain FT (rank=8) | 60.35 | 90.74 | 92.38 | 79.42 | 94.15 | 86.53 | 92.40 | 91.27 | 85.91 |

# H   Experiment Configurations

We provide the source code for reproducing the experimental results reported in the paper. We also summarize some specific configurations that enhance reproducibility.

- **Datasets.** The C4 dataset is publicly available on Huggingface and can be loaded using datasets package `https://github.com/huggingface/datasets`.
- **Random seed.** For all the main experiments for pretraining, we use random seed 42. For fine-tuning experiments, we use random seed 1234.
- **GPU and runtime.** All experiments are conducted with multiple runs on NVIDIA A100-SXM4-40GB/80GB GPUs. The training time vary from 2 hrs (LLaMA 60M) to 5 days (LLaMA 7B) depending on the model size. For fine-tuning, the runtime is roughly the same as LoRA, which is reported on `https://github.com/microsoft/LoRA/tree/main/examples/NLU/examples/text-classification`.

# I   Broader Impact

The paper proposes a simple strategy that achieves both parameter and memory efficiency of training LLMs. We believe this work would produce positive environmental impact by reducing the energy consumption as well as carbon footprint during pretraining large foundation models.

