# OpenReview forum: "SLTrain: a sparse plus low rank approach for parameter and memory efficient pretraining"
_NeurIPS.cc/2024/Conference — NeurIPS 2024 poster_

### Official Review · Reviewer_AVXH · 2024-07-08

**Soundness:** 4
**Presentation:** 3
**Contribution:** 3
**Rating:** 5
**Confidence:** 4

**Summary:**

In this work, the authors introduce SLTrain, a novel method for pre-training large language models (LLMs) that combines sparse and low-rank matrix structures to enhance parameter and memory efficiency. The low-rank component is learned via matrix factorization, while the sparse component is achieved by uniformly selecting the sparsity support at random and learning only the non-zero entries with the fixed support. This method significantly reduces the memory requirement for LLM pre-training while achieving on-par performance.

**Strengths:**

1. This paper introduce SLTrain, serving as the first method to pre-train LLMs with sparse + low rank matrices.
2. The authors provide extensive experimental results demonstrating the effectiveness of SLTrain across various model sizes, from 60M to 7B parameters. The comparison with state-of-the-art methods such as ReLoRA and GaLore is thorough and highlights SLTrain's advantages in memory reduction and parameter efficiency.
3. The authors provide a solid theoretical motivation for combining sparse and low-rank components. The empirical analysis of singular values and the distribution of residuals supports the feasibility and effectiveness of this approach.

**Weaknesses:**

1. As the model size scales up, the perplexity score gap between SLTrain and Full-Rank increases, whereas GaLore maintains a more consistent performance at scale. Consequently, the reduction in parameters with SLTrain leads to suboptimal performance and potential scalability challenges.
2. From Table 3, we observe that the memory efficiency of SLTrain does not translate into faster training speeds. This is due to the inclusion of sparse components during training. Additionally, I am curious about its efficiency during inference. For inference, we have two options: using sparse and low-rank matrices, which should reduce memory usage but increase inference time, or using a larger dense weight matrix, similar to full-rank inference. Therefore, I am interested in the memory and time requirements for both inference modes.
3. I am curious about the potential of using structural sparse patterns, such as butterfly matrices [1], to enhance training performance and efficiency. These structural sparse matrices can be computed in a more efficient way and has better expressivity.

Reference:
[1] Dao, Tri, et al. "Pixelated butterfly: Simple and efficient sparse training for neural network models." arXiv preprint arXiv:2112.00029 (2021).

**Questions:**

See Weaknesses.

**Limitations:**

See Weaknesses.

---

> ### Author Rebuttal · Authors · 2024-08-07
>
> Thank you for acknowledging the strengths of our work and providing many constructive feedback.
>
>
> **1. (W1) As the model size scales up, the perplexity score gap between SLTrain and Full-Rank increases, whereas GaLore maintains a more consistent performance at scale. Consequently, the reduction in parameters with SLTrain leads to suboptimal performance and potential scalability challenges.**
>
> In Table 2, the score gaps between SLTrain and Full-Rank  are 0.09, 1.68, 0.62, and 0.58 for 60M, 130M, 350M, and 1B models, respectfully. The corresponding difference between GaLore and Full-Rank scores are 0.82, 1.00, 0.15, and 0.08, respectively. Hence, for both SLTrain and GaLore, the difference from Full-Rank model increases from 60M to 130M and subsequently decreases. This suggests that the general trend is similar for SLTrain and GaLore. Hence, we respectfully disagree with the reviewer's comment.
>
>
> Notably, the gap between SLTrain and full-rank training can be further reduced by increasing the sparsity level from the outset or by continual pretraining with an additional sparse factor. This approach does not introduce significant memory overhead but enhances performance to levels comparable with full-rank training. Evidence for this is presented in Table 5 of the main paper, where increased rank/$\delta$ corresponds to improved performance.
>
> To validate this claim on larger models, we conducted additional experiments, increasing the sparsity $\delta$ from 0.03 to 0.05 for training LLaMA models with 350M and 1B parameters. The results, detailed in Table 2 of the supplementary one-page PDF, demonstrate that increasing $\delta$ to 0.05 reduces the performance gap while maintaining memory efficiency relative to full-rank training. Furthermore, we highlight that larger models allow for a greater increase in $\delta$ due to the more significant memory gap.
>
>
>
> **2. (W2) Comparison of low-rank plus sparse and dense matrix during inference in terms of memory and time.**
>
> In Table 1 of the one-page PDF,  we have included a comparison of SLTrain and full-rank in terms of inference memory and runtime on LLaMA 130M up to 7B (with the same configuration as in the main paper). We can explicitly observe the trade-off in terms of memory and computation. In particular, as the model size increases, the percentage of memory savings becomes more pronounced, while the computational cost increases is less obvious.
>
> Lastly, we would like to highlight that the computational efficiency of SLTrain highly depends on the implementation as well as the associated hardware. Traditional GPUs are usually poorly-designed for unstructured matrix multiplication. Thus to appropriately leverage the GPU power, we first compute $W = BA +S$ and then multiply by the input with dense operations. It should, however, be noted that we only declare a temporary variable for storing $BA + S$ for each linear layer and when it is executed, it will be freed and be replaced by $BA + S$ in the subsequent linear layer. This only results in a slight increase in memory compared to low-rank parameterization. On the other hand, this is still noticeably smaller in memory compared to full-rank parameterization (such as GaLore) where weights of all layers simultaneously occupy memory.
> This is a strategy that balances memory as well as computation. However, we believe we can match the computational efficiency of the dense model by exploiting many recently introduced sparsity-friendly hardware, such as [1]. Such a hardware is known to match the computational efficiency even for unstructured sparsity.
>
>
> [1] Thangarasa, V., Saxena, S., Gupta, A., and Lie, S. Sparse-IFT: Sparse Iso-FLOP Transformations for Maximizing Training Efficiency. In *ICML 2024*.
>
>
>
>
>
> **3. (W3) Potential of using structural sparse patterns.**
>
> Thank you for your suggestion. We also believe this is interesting to explore and have already discussed this possibility in the concluding remarks in the main paper.

---

> > ### Comment · Reviewer_AVXH · 2024-08-08
> > **Response to Rebuttal**
> >
> > Thank you for your rebuttal. Regarding W1, I still observe a non-negligible performance gap between SLTrain and Full Rank pre-training. In Table 2 of the PDF, the perplexity is slightly higher for SLTrain compared to Full Rank, while the memory usage is slightly lower. Given this, I believe that Full Rank pre-training might still be preferred for its better performance. I would expect SLTrain to achieve a similar performance to be truly impressive.
> >
> > Thank you for your responses to W2 and W3.
> >
> > I will maintain my current score.

---

> > > ### Author Response · Authors · 2024-08-10
> > > **Response to further comments**
> > >
> > > Thank you for your detailed feedback. Based on your comments, it appears that the parameter efficiency of SLTrain, a central contribution of our paper (alongside memory efficiency), may not have been fully recognized. We would like to emphasize that parameter efficiency is a key aspect of our approach, as highlighted in both the abstract and introduction, and substantiated through various experiments presented in the main manuscript and supplementary material.
> > >
> > >
> > >
> > > To further substantiate our claims, we have conducted an additional experiment involving the training of LLaMA 350M with an increased $\delta = 0.1$. The results are presented in the table below, where we also include a comparison with GaLore to better illustrate the advantages of SLTrain.
> > >
> > >
> > >
> > >
> > > |  | PPL| Mem | Param Size|
> > > |----------|--------|--------|--------|
> > > | Full-Rank    | 18.80 |59.34G| 368M  |
> > > | GaLore | 18.95 (+0.8\%)  | 58.35G (-1.7\%)  | 368M (-0\%) |
> > > | SLTrain ($\delta=0.05$)    | 19.24 (+2.3\%)  | 58.00G (-2.2\%) | 200M (-45\%) |
> > > | SLTrain ($\delta=0.1$)  | 18.72 (-0.4\%) | 58.25G (-1.8\%) | 215M (-42\%) |
> > >
> > > As shown in the table, SLTrain with $\delta = 0.1$ not only matches but slightly outperforms the Full-Rank model in terms of perplexity, while requiring only 58\% of its parameter size and maintaining memory efficiency. Across all experiments, SLTrain consistently offers the best trade-off between perplexity, memory usage, and parameter size when compared to other baselines.
> > >
> > > We believe that parameter efficiency is particularly critical during post-pretraining stages. While many existing works, such as [1], focus on model pruning after pretraining, SLTrain directly trains a smaller model from the pretraining stage, effectively reducing parameter size from the outset.
> > >
> > > In summary, the results presented in the table not only demonstrate that SLTrain can achieve performance comparable to the Full-Rank model while maintaining memory efficiency, but also highlight the significant parameter savings it offers.
> > >
> > > **We hope this would lead to a re-evaluation of our work.**
> > >
> > >
> > > [1] Li, Y., Yu, Y., Zhang, Q., Liang, C., He, P., Chen, W., and Zhao, T. LoSparse: Structured compression of large language models based on low-rank and sparse approximation. In *ICML 2023*.

---

> > > > ### Author Response · Authors · 2024-08-14
> > > >
> > > > Dear Reviewer AVXH,
> > > >
> > > > We haven't heard back from you after our last response. We would be grateful if you could look at our responses. We believe our responses would have answered all the questions. Hope to see a positive rerating of the submission.
> > > >
> > > > Best,
> > > > Authors

---

### Official Review · Reviewer_Rr3S · 2024-07-11

**Soundness:** 2
**Presentation:** 2
**Contribution:** 3
**Rating:** 5
**Confidence:** 4

**Summary:**

The submission proposes an approach to reduce memory and computational overhead in training large neural networks. It combines sparse training with low-rank adaptations to achieve efficient training without significant performance degradation. The paper includes an evaluation of SLTrain compared to full-rank, low-rank, ReLoRA, and GaLore models across multiple model sizes and tasks of the LLAMA model family.

**Strengths:**

1.	Ablation Study: The paper yields a good overview of the structure of Llama and analyzes the singular spectrum of several layer matrices, to yield a solid motivation for the sparse + low-rank trick
2.	Comparative Analysis: The paper provides a detailed comparison with other methods such as full-rank, low-rank, ReLoRA, and GaLore for Llama
3.	Implementation Details: The methodology is well-documented, with clear descriptions of how parameters and optimizer states are managed.

**Weaknesses:**

1.	Main concern in this work is in the paragraph starting in line 172. The authors aim for low-rank + sparse pretraining of LLMs to save memory. However, their proposed evaluation of a sparse plus low-rank layer is (AB + S)x, where + denotes a sparse matrix add. This implies that the resulting full matrix Y=AB + S is stored in memory, (if only temporarily) which defeats the purpose of a low-rank formulation. The authors mention “gpu-friendliness” as the reason for this choice, mistaking GPU throughput for real efficiency gain.

- In that regard: How exactly is the “actual memory footprint” in paragraph (line 315) measured? Rigorous details are needed here to make the contribution credible.

 2. The paper is very experimental without theoretical justification of the method (which itself is fine), thus an increased focus on implementation details and actual memory consumption of the implementation is expected.

3.	Results are specific for Llama. The authors consider speficifally the Llama model family in this study. Given the computational effort to train LLMs, this is fine, but is should be clearly stated in the scope, which mentions general LLMs. Have the authors reasonable proof that their method, and mostly, the study of the singular spectrum extends to other LLMs?

**Questions:**

See above

**Limitations:**

yes

---

> ### Author Rebuttal · Authors · 2024-08-07
>
> We sincerely thank you for your feedback. We would like to take this opportunity to address your concerns and questions individually. We hope our clarifications would lead to your re-evaluation of the contribution of this work.
>
>
> **1. (W1) (1) $BA + S$ requires full matrix to be stored in memory (if temporarily) which defeats the purpose of low-rank formulation. (2)  How exactly is the “actual memory footprint” in paragraph (line 315) measured?**
>
> (1) First, we would like to reiterate that the choice of computing $BA + S$ first before multiplying $x$ is due to sparse multiplication being poorly-supported in most GPUs, thus resulting in higher computational cost if computed separately. However, we only declare a temporary variable for storing $BA + S$ for each linear layer and when it is executed, it will be freed and be replaced by $BA + S$ in the subsequent linear layer. This only results in a slight increase in memory compared to low-rank parameterization. On the other hand, this is still noticeably smaller in memory compared to full-rank parameterization (such as GaLore) where weights of all layers simultaneously occupy memory.
>
> To verify the above claims, we perform an additional experiment, comparing the actual maximum memory consumption for the proposed SLTrain linear layer (Algorithm 1, $(BA + S)x$) and standard full-rank linear layer ($W x$) and low-rank linear layer ($BA x$) in a feedforward neural network.
> We include the results for both forward and backward pass where we vary the number of layers in Figure 1 of the additional one-page supplementary PDF. Specifically, we set the input, hidden and output size to be $2048$ and $r = 128$ with $\delta = 0.03$.
> From the figure, we observe that as the number of layers increases, the reduction in memory of SLTrain becomes more evident compared to full-rank model. On the other hand, the memory overhead of SLTrain compared to low-rank model is only marginal. In terms of computational cost, we see compared to full-rank model, SLTrain only requires slight computational overhead, which is due to the scatter-adding operation.
>
> Hence, our proposed $BA+S$ modeling and computation is memory efficient, which *does not* defeat the purpose of low-rank formulation.
>
>
>
> (2) For the second question on "actual memory footprint", we measure the max GPU memory allocated (which is the function 'torch.cuda.max\_memory\_allocated'
> ) during pretraining.
>
>
>
>
> **2. (W2) (1) No theoretical justification. (2)  Details on actual memory consumption of the implementation.**
>
> (1) We provide a formal justification as follows.
>
>
> *Theorem.* Consider a matrix $S \in \mathbb R^{n \times n}$ with support $\mathcal{S}$ sampled uniformly at random with probability $\delta \in (0,1)$, i.e., $\mathbb P[(i,j) \in \mathcal{S}] = \delta$, for all $i, j \in [n]$. Suppose $\delta = \Omega(\log n/n) $, then with probability at least $1- \mathcal{O}(1/n)$, $BA + S$ is full rank for arbitrary randomly generated $B \in \mathbb R^{n \times r}, A \in \mathbb R^{r \times n}$ and for any $r \leq n$.
>
>
> This claims that while $BA$ itself is low rank and has limited expressivity, augmenting $BA$ with a uniform-support sparse matrix renders it to become full-rank.  We will include this theorem in our revised paper.
>
>
> (2) As discussed in the previous point (W1), we measure the max GPU memory allocated (which is the function 'torch.cuda.max\_memory\_allocated') during pretraining. As part of supplementary material to the original submission, we also provide the codes that we use to perform those computations. We will include these details in the revised manuscript.
>
>
>
> **3. (W3) Does the study of singular spectrum extend to other LLMs?**
>
> Yes, we believe the observations on singular spectrum also extend to other LLMs. To validate our conjecture, we repeat the analysis of singular spectrum for pretrained Pythia 70M, downloaded from Hugging Face. Specifically, we set the rank $r = 128$ and extract the best rank $r$ approximation of the learned weight matrices.
> The results are shown in Figure 2 of the supplementary one-page PDF, where we observe that the residual after removing the best low-rank approximation vary smoothly and has small magnitude.

---

> > ### Comment · Reviewer_Rr3S · 2024-08-09
> >
> > Thank you for the answer and clarifying remarks to point 2) and 3).
> >
> > ad 1): The argument for the memory efficiency of the method is that the full weight matrix is only stored temporarily. Can the authors comment on the parallelizability constraints that this temporal full-matrix constraints bring?
> > Specifically: The method seemingly generates the full matrix during the layer evaluation of layer $k$, then discards it again to evaluate layer $k+1$? If so, can the authors elaborate how the gradient tape is stored? It must not store the temporary (AB + S) matrix, since then one would observe similar memory footprint as the full model.
> > Second, can the authors elaborate on the practicability of combining their approach with, e.g. pipelining, where multiple layers are evaluated in parallel? Is their method still applicable, since in this scenario the temporarily stored AB+S needs to be constructed in many layers simultaneously, potentially leading to memory spikes.

---

> > > ### Comment · Reviewer_Rr3S · 2024-08-09
> > >
> > > Out of curiosity: Can the authors elaborate a bit on the choice of the hyperparameters, specifically the sparsity parameter $\delta$ and rank $r$. How can one choose them in practise, and have they observed how they influence each other, e.g. a higher sparsity score means that the models require less rank, or something similar.

---

> > > > ### Author Response · Authors · 2024-08-10
> > > > **Response to further questions**
> > > >
> > > > Thank you for the follow-up questions. Please see our responses below.
> > > >
> > > >
> > > > **1. On gradient memory of $BA + S$.**
> > > >
> > > > No, we never need to store the temporary $(BA + S)$ matrix for gradient computation and this is one of the main points we highlight in Section 3.2 (Line 181 of the main paper). We only need to store the necessary components in order to compute the gradients, i.e., $B, A$ and the indices and values for $S$. We have defined a customized layer specifically for this purpose. Please refer to Algorithm 1 and provided codes in supplementary for more details.
> > > >
> > > >
> > > >
> > > > **2. On combining SLTrain with pipelining.**
> > > >
> > > > We have not tried out such pipelined scenarios in the paper. However, a straightforward combination of SLTrain with pipelining may not yield memory savings. For such settings, we would consider the approach of computing $BAx$ and $Sx$ separately albeit with a trade-off on computational cost.
> > > >
> > > > In summary, while our method is applicable in pipelined scenarios, achieving the full efficiency gains may require additional memory management strategies when layers are evaluated in parallel.
> > > >
> > > >
> > > >
> > > >
> > > >
> > > >
> > > > **3. On choice of hyperparameters.**
> > > >
> > > > In practice, we often can try out different combinations to understand the trade-offs better. Please refer to Table 5 of the main paper, where we compare the model's performance across various choices of sparsity parameter $\delta$ and rank $r$. Generally, a higher sparsity parameter allows the model to achieve comparable performance with a lower rank, as you correctly noted.

---

> > > > > ### Author Response · Authors · 2024-08-14
> > > > >
> > > > > Dear Reviewer Rr3S,
> > > > >
> > > > > We haven't heard back from you after our last response. We would be grateful if you could look at our responses. We believe our responses would have answered all the questions. Hope to see a positive rerating of the submission.
> > > > >
> > > > > Best,
> > > > > Authors

---

### Official Review · Reviewer_Wf5L · 2024-07-12

**Soundness:** 3
**Presentation:** 3
**Contribution:** 3
**Rating:** 6
**Confidence:** 3

**Summary:**

The authors propose SLTrain that performs a low-rank factorization of the weights as well as a sparse matrix of factors that represents which parameters to update. The authors show that their method can achieve significant memory savings compared to GaLore while retaining performance.

**Strengths:**

- the paper is well written and easy to read
- the method is very simple to integrate to the model as randomly generating sparse factors is easy to initialize
- the savings look significant compared to GaLore

**Weaknesses:**

- no strong theoretical justification why the sparse method works.
- it is not clear how much sparsity is needed for a specific task. Is the degree of sparsity impactful on the performance?
- what if we only parametrize the weights with BA and not BA+S, wouldn't that decrease the number of parameters? and how well will it perform?
- wouldn't it be more useful if we regenerate the random sparse factors every couple of iterations to have a more effective pretraining?
- how much do the results depend on the randomness of the sparse factors? can we have multiple runs and a standard deviation to see if that randomness play a big role in the results?
- comparing GaLore and SLTrain is not really apple-to-apple as GaLore is based on the gradients and SLTrain is based on the weights. What if you had GaLore+SLTrain where gradients and weights are both made more efficient?

**Questions:**

Please address the weaknesses above

**Limitations:**

yes

---

> ### Author Rebuttal · Authors · 2024-08-07
>
> Thank you for the positive and constructive comments on our work.
>
> **1. (W1) Lack of theoretical justification.**
>
> We motivate the low-rank plus sparse modelling from the empirical observations (Figure 2 in the main text and Figure 2 in the one-page supplementary PDF) that the pretrained weights of LLMs can be well-approximated by a low-rank and a (uniformly) sparse factor. To further justify the modelling, we provide a formal justification as follows.
>
> *Theorem.* Consider a matrix $S \in \mathbb R^{n \times n}$ with support $\mathcal{S}$ sampled uniformly at random with probability $\delta \in (0,1)$, i.e., $\mathbb P[(i,j) \in \mathcal{S}] = \delta$, for all $i, j \in [n]$. Suppose $\delta = \Omega(\log n/n) $, then with probability at least $1- \mathcal{O}(1/n)$, $BA + S$ is full rank for arbitrary randomly generated $B \in \mathbb R^{n \times r}, A \in \mathbb R^{r \times n}$ and for any $r \leq n$.
>
> This suggests that although $BA$ itself is low-rank, which has limited expressivity, $BA + S$ is full-rank with high probability as long as the support is selected uniformly at random with sufficient non-zero entries. We will include this theorem in our revised draft.
>
>
>
>
> **2. (W2) Is the degree of sparsity impactful on the performance?**
>
> Yes, the degree of sparsity would impact the performance. We have already shown such results in Table 5 in the main paper where we vary the degree of sparsity ($\delta$). A trade-off exists between the performance and memory consumption, where more parameters (higher memory consumption) in general correspond to better performance.
>
> **3. (W3) What if we only parametrize the weights with BA and not BA+S, wouldn't that decrease the number of parameters? and how well will it perform?**
>
> The low-rank only approach ($BA$) does not work well in pretraining, e.g., already noted in [59]. In fact that is the motivation for our work (we mention this in Lines 64-65) and that full-rankness is required for effective pretraining. Indeed, $BA+S$ parameterization leads to full-rank weights (as our above theorem suggests) and consistently outperforms $BA$ in scores. See the comparison in Table 2 of main paper, where the baseline "Low-Rank" refers to $BA$ parameterization.
>
>
>
> **4. (W4) More useful when random sparse factors are regenerated.**
>
> We are not sure if regenerating random sparse factors would be useful. Our experiments show that learning of the sparse component $S$ with a fixed-random support is as useful as learning the low-rank part. If we were to regenerate the mask every couple of iterations, then it would mean that we are ignoring the factor learnt $S$ till now and only leveraging the learnt $BA$, which may lead to a decrease in performance. Furthermore, our experiments in Figure 4 suggest that the results remain almost the same with different uniform masks.
>
>
>
> **5. (W5) How much do the results depend on the randomness of the sparse factors? can we have multiple runs and a standard deviation to see if that randomness play a big role in the results?**
>
> We have already done this. We have validated the influence of randomness for 60M and 130M model by running the models 5 times and presented the result in Figure 4 of the main paper that the randomness does not significantly affect the performance of the two models. In particular, the perplexity for 60M model at 1.1B is 33.91 with a standard deviation of 0.18 and the perplexity for 130M model at 2.2B is 26.01 with a standard deviation of 0.10.
>
> **6. (W6) Comparing GaLore and SLTrain is not really apple-to-apple as GaLore is based on the gradients and SLTrain is based on the weights. What if you had GaLore+SLTrain where gradients and weights are both made more efficient?.**
>
>
> First, we highlight that both GaLore and SLTrain aim to do memory efficient pretraining but using different strategies. Therefore, it is not  unfair to compare them.
>
> On your suggestion about GaLore+SLTrain, indeed, we have already discussed this possibility explicitly in Line 217, given our proposed strategy is orthogonal to the development of GaLore. However, we believe that this is out of scope for the current submission.

---

> > ### Comment · Reviewer_Wf5L · 2024-08-14
> >
> > Thank you for the rebuttal and for addressing most of my concerns. I have increased my score by 1.

---

### Author Rebuttal · Authors · 2024-08-07

Dear Reviewers and ACs

We sincerely appreciate the time and effort you have invested in managing our submitted paper. We are especially grateful for your constructive and thoughtful feedback. In response to your comments, we have provided a formal justification and we have also included a *one-page supplementary PDF* that contains the following:


- Figure 1: Comparison of memory and runtime between full-rank linear layer and SLTrain linear layer

- Table 1: Comparison of inference memory and runtime between full-rank and SLTrain on LLaMA models

- Figure 2: Illustration of pretrained weight decomposition of Pythia 70M model.

- Table 2: Perplexity and actual max memory allocated for pretraining LLaMA 350M and 1B models with increased $\delta = 0.05$.

- Table 3: Perplexity comparisons of pretraining LLaMA 7B model to 5.2B training tokens.

We hope that our responses and additional experimental results adequately address all your questions and concerns. If there are any additional areas that require further clarification or improvement, we would be more than willing to make the necessary adjustments.

Regards

Authors

---

### Author Response · Authors · 2024-08-13

We sincerely thank the reviewers for their constructive feedback, which has significantly strengthened our paper. We believe we have effectively addressed all concerns and questions raised thus far. Specifically:


- We have provided a theoretical justification for the low-rank plus sparse modeling in response to the concerns raised by Reviewer Wf5L and Reviewer Rr3S.


- We have offered detailed explanations and numerical comparisons to clarify how $BA + S$ achieves memory efficiency, addressing the primary concern of Reviewer Rr3S.


- We have conducted additional experiments demonstrating that SLTrain can achieve comparable performance to full-rank models while significantly reducing parameter size and maintaining memory efficiency, addressing Reviewer AVXH’s concerns regarding the performance gap between SLTrain and full-rank models.


- We have emphasized the parameter efficiency of the proposed model, alongside its memory efficiency, one of the key contributions that tend be overlooked by the reviewers.


- We have addressed all other questions raised by the reviewers.


As the discussion period deadline approaches, we would greatly appreciate a **reconsideration of the scores** if there are no further questions. However, if any additional concerns arise, we remain fully committed to addressing them promptly.

---

### Decision · Program_Chairs · 2024-09-25

**Decision:**

Accept (poster)

**Comment:**

There was a disagreement among reviewers on this paper. However, the author pointed out a significant mis-understanding of the dissenting reviewer. Please clarify this issue in the next version.